# The Interaction of Human and *Epstein–Barr Virus* miRNAs with Multiple Sclerosis Risk Loci

**DOI:** 10.3390/ijms22062927

**Published:** 2021-03-13

**Authors:** Ali Afrasiabi, Nicole L. Fewings, Stephen D. Schibeci, Jeremy T. Keane, David R. Booth, Grant P. Parnell, Sanjay Swaminathan

**Affiliations:** 1Systems Biology and Health Data Analytics Lab, The Graduate School of Biomedical Engineering, UNSW Sydney, Sydney, NSW 2052, Australia; ali.afrasiabi@sydney.edu.au; 2EBV Molecular Lab, Centre for Immunology and Allergy Research, Westmead Institute for Medical Research, University of Sydney, Westmead, NSW 2145, Australia; nicole.fewings@sydney.edu.au (N.L.F.); stephen.schibeci@sydney.edu.au (S.D.S.); jeremy.keane@sydney.edu.au (J.T.K.); 3Department of Medicine, Western Sydney University, Sydney, NSW 2560, Australia

**Keywords:** MS, EBV, GWAS, miRNA, *EBNA2*, *ZC3HAV1*

## Abstract

Although the causes of Multiple Sclerosis (MS) still remain largely unknown, multiple lines of evidence suggest that *Epstein–Barr virus* (EBV) infection may contribute to the development of MS. Here, we aimed to identify the potential contribution of EBV-encoded and host cellular miRNAs to MS pathogenesis. We identified differentially expressed host miRNAs in EBV infected B cells (LCLs) and putative host/EBV miRNA interactions with MS risk loci. We estimated the genotype effect of MS risk loci on the identified putative miRNA:mRNA interactions in silico. We found that the protective allele of MS risk SNP rs4808760 reduces the expression of *hsa-mir-3188-3p*. In addition, our analysis suggests that *hsa-let-7b-5p* may interact with *ZC3HAV1* differently in LCLs compared to B cells. In vitro assays indicated that the protective allele of MS risk SNP rs10271373 increases *ZC3HAV1* expression in LCLs, but not in B cells. The higher expression for the protective allele in LCLs is consistent with increased *IFN* response via *ZC3HAV1* and so decreased immune evasion by EBV. Taken together, this provides evidence that EBV infection dysregulates the B cell miRNA machinery, including MS risk miRNAs, which may contribute to MS pathogenesis via interaction with MS risk genes either directly or indirectly.

## 1. Introduction

Multiple Sclerosis (MS) is a disabling neurological disease that disproportionally affects young people [1]. The causes of MS remain largely unknown, although recent genome-wide association studies (GWAS) have strongly implicated > 200 MS risk Single Nucleotide Polymorphisms (SNPs) associated with immune-related genes that may play an important part in pathogenesis [2]. In recent years, there has been increasing evidence that viruses, particularly *Epstein–Barr virus* (EBV), are associated with the development of MS [3]. EBV infects and drives B cells from a resting state into an activated B blast cell, which eventually can become an immortalized latency III infected resting memory B cell [4]. Virtually all patients with MS are seropositive for EBV [5,6], whilst 94% of the general population are positive [7]. Of more relevance, late infection or severe infection with EBV can increase the risk of developing MS by 15-fold, an increase not found with other viral pathogens [7]. In addition, other studies have shown that patients who are seronegative to EBV but then go on to develop MS in later life become seropositive to EBV prior to diagnosis of MS, suggesting exposure to EBV may play an important role in developing MS [8]. Several mechanisms have been proposed to explain how EBV may lead to MS, including EBV cross-reactivity, EBV bystander damage, the mistaken self-hypothesis, and the EBV infected autoreactive B cell hypothesis [9].

We reasoned that the variability in host cellular response to EBV infection might define an individuals’ MS susceptibility. Thus, mapping the crosstalk between EBV infection molecular pathways and MS risk loci can be used as a handle to identify those differences in host response to EBV infection that may facilitate MS development, and will provide insight into potential therapeutic targets for MS.

In recent years, microRNAs (miRNAs) have emerged as important regulators of gene expression [10]. miRNAs are small RNA species, 19–22 nucleotides in length, that interact with mRNAs via a protein complex at the 3′ untranslated region (3′UTR) [10]. There are 44 EBV encoded viral miRNAs, which EBV uses to downregulate expression of its own proteins as well as that of the host to minimize exposure to the host immune response [11]. It has been postulated that miRNAs, within the EBV infected B cell context, may contribute to MS pathogenesis via two potential mechanisms: (1) EBV miRNAs may target MS risk genes in the process of B cell transformation to increase risk of MS, and/or (2) host cellular miRNAs may be dysregulated by EBV infection and consequently, these dysregulated host miRNAs alter B cell function [12].

To interrogate the role of EBV miRNAs and host miRNAs dysregulated due to EBV infection in MS pathogenesis, we first identified the differentially expressed miRNAs between the EBV infected B cells, referred to as lymphoblastoid cell lines (LCLs), and B cells. Then, we interrogated the following possible interactions between miRNAs within LCLs (EBV and host cellular miRNAs) which could be involved in MS pathogenesis: (I) Host MS risk miRNAs may target MS risk genes (II) host miRNA binding sites among MS risk genes may be affected by MS risk SNPs (III) EBV miRNAs may target MS risk genes (IV) EBV miRNA binding sites among host MS risk genes may be affected by MS risk SNPs (Figure 1). From the first two, we identified those dysregulated host miRNAs that may contribute to MS pathogenesis. Further, the last two led us to propose a novel approach for targeting EBV, with the aim of slowing MS disease development and/or progression.

Taken together, we here provide supporting evidence that EBV miRNAs and host B cell miRNAs dysregulated upon EBV infection contribute to MS pathogenesis via interaction with MS risk genes both directly and indirectly.

## 2. Results

### 2.1. MS Risk miRNAs Are Differentially Expressed between LCLs and B Cells

We designated host miRNAs as MS risk miRNAs if they were If they were the closest miRNA upstream or downstream of an MS risk SNP (within 1 Mega base pairs) or were an expression quantitative trait locus (eQTL) with MS risk SNPs or SNPs in linkage disequilibrium (LD) with MS risk SNPs. We identified 12 miRNAs proximal to MS risk SNPs which encode 19 mature miRNAs (Appendix A). None of these miRNAs were in eQTL with their proximal MS risk SNPs at an FDR of less than 5%. However, 10 out of 24 SNPs in LD with MS risk SNP rs4808760 (R^2^ > 0.8) were in eQTL with *hsa-mir-3188-3p* at *p* value less than 10^−6^. The risk allele of rs4808760 increased the expression of *hsa-mir-3188-3p*. Interestingly, this LD block is targeted by the EBV transcription factor *EBNA2* (Figure 2 and Appendix A). In total, we identified 20 MS candidate risk miRNAs (Table 1).

We then determined those MS risk miRNAs which are differentially expressed between LCLs and B cells. From 1012 detected host miRNAs, 135 mature miRNAs were differentially expressed between LCLs and B cells with a *p* value of less than 0.05. Of those, 35 miRNAs were downregulated, and 100 miRNAs were upregulated in LCLs compared to B cells. Of the 20 identified MS risk miRNAs, nine miRNAs were detected in both LCLs and B cells. Furthermore, we found that five miRNAs including one MS risk miRNA (*hsa-miR-4435*) were expressed only in B cells but not in LCLs, and 14 miRNAs including one MS risk miRNA (*hsa-miR-4492*) were expressed only in LCLs, but not in B cells. (Table 2 and Appendix A).

MS risk miRNAs were over-represented in the differentially expressed miRNAs (*n* = 6, representation factor: 5, *p* < 10^−4^). Four MS risk miRNAs were upregulated, while two were downregulated in LCLs compared to B cells (Figure 3). MS risk miRNAs were mainly overexpressed in LCLs compared to B cells (*n* = 4, representation factor: 4.5, *p* < 0.008). This suggests that EBV infection results in an altered miRNA profile in B cells as well as dysregulation of MS risk miRNAs. Alteration in the B cell miRNA profile may facilitate the establishment/maintenance of the EBV infection. Although not all the altered miRNAs are MS risk miRNAs, they may contribute to MS pathogenesis by affecting MS risk genes either directly or indirectly.

### 2.2. Which MS Risk Genes Are Targeted by MS Risk miRNAs?

PAR-CLIP data were available for five MS risk miRNAs; *hsa-let-7a-5p*, *hsa-miR-125b-5p*, *hsa-miR-21-5p*, *hsa-miR-21-3p* and *hsa-mir-3188-3p*. The putative targets for these miRNAs were identified via pipeline A (Appendix A). Interestingly, several MS risk genes were identified as being targeted by MS risk miRNAs *hsa-let-7a-5p*, *hsa-miR-21-5p* and *hsa-miR-125b-5p* (Table 3, Appendix A).

Since miRNAs reduce the expression of their targeted mRNAs [13], a negative correlation between the expression of miRNA and mRNA is expected. However, most of the identified miRNA:mRNA duplexes had an non-significant or significant positive correlation in the Geuvadis LCL cohort (Appendix A). We then assessed the correlation between these miRNAs and the expression of their targeted mRNAs in LCLs and B cells. Most of the miRNA:mRNA duplexes had non-significant correlations in either the LCL or B cell contexts. However, significant negative correlations were found between *hsa-let-7a-5p* and *UFL1*, *PDCD4*, *RBM12B*, *PAPD4*, *IFI44L*, *COL19A1*, and *ZNF460* in the B cell context (Rho = −1, *p* value 0.0167, FDR 0.47), but not significant in the LCL context (Appendix A). This suggests these interactions may be disrupted upon EBV infection and the subsequent transformation of B cells to LCLs.

To determine the effect of the MS risk genotype on the predicted miRNA:mRNA interactions, we calculated the correlation between miRNA expressions and their putative targets in the Geuvadis LCL cohort (Euro samples *n* = 373) for all genotypes (whole population), risk (homozygote for risk allele), heterozygote (heterozygote for risk and protective alleles), and protective (homozygote for protective allele) genotypes of proximal MS risk SNPs associated with these MS risk miRNAs. The correlation network in different MS risk genotype conditions for miRNA:mRNA is described in detail in Appendix A.

From 204 putative interactions between MS risk miRNAs and their targeted mRNAs, 20 were correlated significantly including one MS risk gene *ZFP36L1* which is targeted by MS risk miRNA *hsa-miR-21-5p*. Five of the interactions were associated with MS risk SNP genotypes, however none were MS risk genes (Table 4).

### 2.3. Which EBV miRNAs Target the Host MS Risk Genes?

Analysis pipeline A predicted interaction between five EBV miRNAs and 13 MS risk genes (Figure 4A). *BART15* targets *JARID2*, *AFF1*, *SLAMF1*, *LPP* and *ATXN1*. *BART1-3p* targets *RGS1*, *CLECD2* and *FAM76B*. *BART2-3p* targets *AFF1*, *ETS1* and *GRB2*. *BART4-5p* targets *RCOR1* and *CD200R1*. *BHRF1-2-5p* targets *MALT1* and *EST1*. It is noteworthy that the MS risk SNPs associated with *CLECD2*, *AFF1*, *RCOR1*, *MALT1*, *CD200R1*, *ETS1* and *SLAMF1* MS risk genes are targeted by *EBNA2* [14]. The expression of *CD200R1*, *MALT1*, *RCOR1* and *AFF1* was associated with MS risk genotype in the LCL context (Figure 4B), suggesting these genotype effects could be due to EBV miRNA interaction with these genes. The binding sites of the above-mentioned EBV miRNAs on their targeted genes are described in detail in Appendix A.

Moreover, the genotype effect of MS risk SNPs on these genes which are targeted by EBV miRNA were tested by calculating the correlation between EBV miRNA expression and their targeted MS risk genes in the whole population, risk, heterozygote, and protective genotype conditions in the Geuvadis Euro LCL cohort (*n* = 373). None of the interactions were negatively correlated in the whole population, indicating that EBV miRNAs may not interact with MS risk genes in a genotype dependent manner. However, two significant positive correlations were found in genotype groups analysis, for *BART1-3p* and *FAM76B* (Rho = 0.15, *p* value 0.024, FDR 0.46) for the risk genotype group, and for *BHRF1-2-5p* and *MALT1* (Rho = 0.14, *p* value 0.03, FDR 0.46) for the protective genotype group (Appendix A).

### 2.4. Which Host and EBV miRNA Binding Sites among Host MS Risk Genes Are Affected by MS Risk SNP Genotype?

Although none of the MS risk SNPs were located within the putative EBV miRNA binding sites, five SNPs in LD with MS risk SNPs were located within the putative miRNA binding sites on *MALT1*, *IKZF3*, *LBH*, *ZC3HAV1* and *ZFP36L1* (Figure 5).

There were no binding sites for EBV miRNAs proximal to these SNPs. However, we found eight binding sites for host miRNAs: three on *ZC3HAV1* of which one was shared with *hsa-miR-98-5p*, *let-7i-5p*, *let-7b-5p*, *let-7c-5p*, *let-7d-5p*, *let-7e-5p*, *let-7g-5p* and *let-7a-2-5p* (Figure 6A) and two independent binding sites for *hsa-miR-196b-5p* and *hsa-miR-149-3p*, two independent binding sites on *ZFP36L1* for *hsa-miR-3190-3p* and *hsa-miR-629-5p*, one binding site on *MALT1* for *hsa-miR-33a-3p*, one binding site on *LBH* which is shared for *hsa-miR-7-1-5p*, *hsa-miR-7-2-5p* and *hsa-miR-7-3-5p*, and one binding site on *IKZF3* for *hsa-miR-1306-5p* (Appendix A). To evaluate whether these putative interactions are functional in the LCL context, we compared the correlation of expression between putative miRNAs and their targeted genes in the B cell and LCL contexts (Appendix A). Although no significant negative correlation between putative miRNAs and their targeted genes in the LCL context was found, a significant negative correlation (Rho = −1, *p* value 0.016, FDR 0.2) was found between *hsa-let-7b-5p* and *ZC3HAV1* in the B cell context. *hsa-let-7b-5p* is one of the host miRNAs which is differentially expressed between LCLs and B cells and its expression is lower in the LCL context compared to B cells (*p* value 0.002 and fold change 0.17) (Appendix A). Correlation network analysis of the European population of the Geuvadis LCL cohort (*n* = 373) showed a significant positive correlation between *hsa-let-7b-5p* and *ZC3HAV1* (Rho = 0.09663, *p* value 0.0407, FDR 0.26) in the whole population. In addition, a significant negative correlation was found between *hsa-let-7i-5p* and *ZC3HAV1* (Rho = −0.2731, *p* value 0.0125, FDR 0.13) in the risk genotype group of the risk MS SNP for *ZC3HAV1* (Appendix A).

Furthermore, A significant positive correlation was found between *hsa-miR-1306-5p* and *IKZF3* in the whole population (Rho = 0.09354, *p* value 0.0476, FDR 0.26). A significant negative correlation (Rho = −0.2095, *p* value < 0.00001, FDR 0.00056) was found between *ZFP36L1* and *hsa-miR-629-5p* in the whole population. Although only the correlation between expression of *hsa-miR-7-1-5p* and *LBH* was slightly negatively significant in the whole population (Rho = −0.09 and *p* value 0.04, FDR 0.26), *hsa-mir-7-1-5p*, *hsa-mir-7-2-5p* and *hsa-mir-7-3-5p* were significantly negatively correlated with *LBH* expression level (*p* value 0.01) in the risk genotype group but not in the protective genotype group (Appendix A).

Taken together, this indicates that the LCL miRNA machinery interaction with MS risk genes *ZFP36L1*, *LBH*, and *ZC3HAV1* is a stable interaction. Additionally, this interaction appears stronger in carriers of the risk allele genotype for *LBH*.

### 2.5. Which Host and EBV miRNA Binding Sites among Host MS Risk Genes Are Affected by MS Risk SNP Genotype?

The miRNA target site analysis indicated that several miRNAs potentially targeted the MS risk gene *ZC3HAV1*. From these, *hsa-let-7b-5p* was correlated with *ZC3HAV1* negatively in B cells, and positively in LCLs. In addition, *hsa-let-7b-5p* was significantly down-regulated in LCLs compared to B cells. Based on this, we reasoned that the regulation of *ZC3HAV1* may be different in the LCL context compared to that of B cells. Therefore, we assessed the genotype effect of the MS risk SNP rs10271373 and its associated gene *ZC3HAV1* in LCLs and B cells in vitro. *ZC3HAV1* expression was associated with the rs10271373 MS risk SNP in the LCL context, with the risk allele reducing the expression of *ZC3HAV1* (*p* value 0.01), but such an association was not observed in the B cell context (*p* value 0.78). This LCL-specific MS risk SNP genotype effect on *ZC3HAV1* expression may be due to the interaction of *hsa-let-7b-5p* with *ZC3HAV1* (Figure 6B).

## 3. Discussion

Here, we aimed to identify the potential contributory EBV and host cellular miRNAs by which EBV infection affects MS pathogenesis. We identified the miRNA expression profile of B cells before and after EBV transformation. The expression of several cellular miRNAs within B cells was altered following transformation into LCLs. Our results showed that some miRNAs were expressed only in uninfected B cells and were completely suppressed in the LCL context. On the other hand, some miRNAs are solely expressed in LCLs and not expressed in uninfected B cells. Such unique differences in the miRNA expression profiles of LCLs and B cells suggests that the B cell miRNA profile changes with the B cell phenotype. Many of these miRNAs which were exclusively expressed in LCLs but not in B cells are involved in regulating processes involved in cell proliferation, while the ones that were suppressed completely had an inhibitory effect on cell proliferation [15]. Almost half of the MS risk miRNAs were expressed in the B cell and LCL contexts. More MS risk miRNAs were differentially expressed between LCLs and B cells more than would be expected by chance, and the majority were upregulated in LCLs compared to B cells. We also found that the expression of MS risk miRNA *hsa-miR-4492* is LCL context-specific and is not expressed in resting B cells, whereas MS risk miRNA *hsa-miR-4435* expression is B cell context-specific and not expressed in LCLs. These pieces of evidence suggest that EBV not only produces its own miRNAs but also alters the host B cell miRNA profile, including MS risk miRNA expression.

To determine highly accurate target sites of host and EBV miRNAs, we used the most efficient and reliable miRNA target site prediction tools, TargetScan [16] and miRmap [17] for host miRNAs, and RepTar [18] for EBV miRNAs. We then cross-validated the predicted miRNA:mRNA duplexes with the publicly available photoactivatable-ribonucleoside enhance crosslinking and immunoprecipitation (PAR-CLIP) based data in 6 different LCLs [19]. PAR-CLIP is a high resolution experimental method for determining putative miRNA:mRNA duplexes. Our analysis showed that MS risk miRNAs potentially interact with several genes including MS risk genes.

Given the fact that miRNAs usually degrade their targeted mRNAs resulting in reduced mRNA expression [13], we reasoned that the significance of a negative correlation between a miRNA and its targeted mRNA expression indicates the stability and accuracy of the putative miRNA:mRNA duplexes. Thus, we assessed the correlation between expression of each miRNA and their putative targets in the LCL and B cell contexts. Further, we assessed the MS risk genotype effect on the validated miRNA:mRNA interactions by constructing a correlation network for candidate miRNAs and targeted mRNA expression in B cells and LCLs. However, most of the identified miRNA:mRNA duplexes had a non-significant or positively significant correlation in the Geuvadis LCLs. The interaction between miRNA and mRNA is a dynamic event and these interactions can be functional at a specific time point or under specific conditions. Thus, it may possible that those predicted interactions with no correlation or positive correlation are functional in different stages of EBV infection/transformation of B cells. To confirm the accuracy of these interactions the expression of miRNA and mRNA at different time points of the EBV life cycle in B cells needs to be assayed.

Overall, we identified interactions that were associated with MS risk genotype as well as those which were more associated with either B cell or LCL contexts. Taken together, we concluded that EBV may establish this new miRNA profile in transformed B cells, with a unique repertoire of cellular EBV-encoded miRNAs, to maintain its lifecycle by regulating B cell genes as well as preventing necessary miRNA:mRNA interactions involved in B cell homeostasis related molecular pathways. This process may contribute to MS pathogenesis by affecting MS risk genes, either directly or indirectly.

The expression of one MS risk miRNA, *hsa-mir-3188-3p*, was associated with the risk SNP rs4808760 in LCLs, where the protective allele reduced *hsa-mir-3188-3p* expression. This genotype effect on the expression was consistent across most of the SNPs in LD with risk SNP rs4808760, supporting the stability of this association. Furthermore, *EBNA2* interacts with the rs4808760 LD block, highlighting the importance of *hsa-mir-3188-3p* in the EBV life cycle. To our knowledge there is no data about the function of this miRNA in either B cells or LCLs. However, it has been reported that *hsa-mir-3188-3p* decreases the proliferation of the EBV-positive nasopharyngeal carcinoma cell line (HONE1-EBV) by interacting with *mTOR* [20]. Altogether, this suggests that this MS risk miRNA may also play a role in maintenance of EBV-infected B cell proliferation. It is notable though that the protective allele reduced *hsa-mir-3188-3p* expression. The function of *hsa-mir-3188-3p* in the LCL context needs to be determined to interpret the protective allele effect on EBV life cycle processes.

Our results showed that several MS risk genes were potentially targeted by EBV miRNAs. We previously showed that four of these MS risk genes (*MALT1*, *RCOR1*, *AFF1* and *CD200R1*) were expressed in an MS risk SNP genotype-dependent manner and were more associated with the LCL context than that of other immune cells [14]. Furthermore, we have shown that *EBNA2* binds at the associated MS risk SNPs of these four genes [14]. Therefore, EBV may use two independent regulatory mechanisms to regulate these MS risk genes, mediated by *EBNA2* and by EBV-encoded miRNAs, indicating the importance of these genes in the EBV life cycle as well as the link between EBV infection and MS genetic risk factors. Aligned with our results, the interaction between *BHRF1-2-5p* and *MALT1* has been confirmed using a luciferase assay [15,21]. *MALT1* plays a key role in B cell proliferation by modulating the *NF-κB* pathway. However, the role of other MS risk genes potentially targeted by EBV miRNAs in the EBV life cycle needs to be studied. Moreover, it is worth mentioning that EBV B95.8 strain, the most common EBV lab model, expresses a small proportion of EBV derived miRNAs compared to other EBV strains due to a big deletion in the genomic region containing the *BART* family miRNAs. There is a possibility that EBV miRNAs other than these 13 expressing miRNAs in B95.8 strain may be involved in MS pathogenesis. Thus, to identify the role of all EBV miRNAs in MS pathogenesis, PAR-CLIP, small RNAseq and RNAseq data from LCLs transformed by other EBV strains with a complete miRNA profile would be needed.

It has been shown that the EBV transcription factor *EBNA1* alters the expression of cellular miRNAs, notably by upregulating *let-7* family miRNAs to maintain EBV latency in nasopharyngeal carcinoma (NPC) cell lines [22]. In contrast, our results indicated that two *let-7* miRNAs, *hsa-let-7b-5p* and *hsa-let-7e-5p*, were downregulated in LCLs compared to B cells. EBV positive NPC cell lines and LCLs are in latency phase I and III, respectively. We then reasoned that the *let-7* family miRNA expression is altered by the EBV latency phases, which indicates this miRNA family may play a critical role in EBV phase switching and maintenance of the EBV life cycle stages. One MS risk gene, *ZC3HAV1*, had six putative target sites among the *let-7* miRNA family in its 3′UTR, adjacent to one SNP (rs10250457) that is in LD with the MS risk SNP rs10271373. Among these, *hsa-let-7b-5p* expression was negatively correlated with *ZC3HAV1* in B cells, but positively correlated in LCLs. The risk gene *ZC3HAV1* is known to be used by the host to amplify the interferon response [23]. Therefore, the higher expression of *ZC3HAV1* for the protective allele in LCLs is consistent with an increased *IFN* response and so decreased immune evasion by EBV.

We have previously shown that the expression of EBV miRNAs *BART4-3p* and *BART3-5p* are both associated with the MS risk gene *PVR* [24]. Here, we showed that EBV infection dysregulates the B cell miRNA machinery, including MS risk miRNAs. Furthermore, we provided additional evidence that other MS risk genes *ZC3HAV1*, *MALT1*, *ZFP36L1*, *LBH*, *RCOR1*, *AFF1* and *CD200R1* may be regulated by EBV infection through targeting by EBV miRNA and EBV-driven dysregulation of cellular miRNAs. Together, these lines of evidence suggest that the EBV life cycle is associated with MS risk gene function. This may explain one of the molecular mechanisms responsible for the high EBV seropositivity in MS patients. However, a key question remains for further investigations: do MS risk loci contribute to pathogenesis through direct interactions with EBV elements or do MS and EBV simply coincidently both interact at MS risk loci, which means that EBV may not directly drive MS pathogenesis.

In conclusion, this study provides evidence that EBV infection dysregulates the B cell miRNA machinery, including MS risk miRNAs, which may contribute to MS pathogenesis via interaction with MS risk genes either directly or indirectly and provides a possible therapeutic target for the treatment for multiple sclerosis.

## 4. Materials and Methods

### 4.1. Dataset Preparation

The EBV strain B95.8 and human miRNAs sequences were obtained from miRBase 22.0 [25]. The MS risk SNPs were obtained from the most recent MS GWAS meta-analysis [2]. We then calculated SNPs in LD with an R^2^ ≥ 0.8 and 200 kb distance threshold with the risk SNPs in the CEU population using the LDlink R package [26]. This final MS risk SNP list contained 201 non-*HLA* MS risk SNPs and 4707 SNPs in LD with them. The candidate MS risk genes were identified as the closest genes in upstream and downstream of GWAS MS risk SNPs (within 1 Mega base pairs) which reached the genome wide significance cut-off of 5 × 10^−8^. This resulted in 223 candidate MS risk genes.

To determine the MS risk miRNAs, we used the same SNPs set that were utilized for identifying MS risk genes. The closest host miRNAs in upstream or downstream of an MS risk SNP (within 1 Mega base pairs) or were an eQTL with MS risk SNPs or SNPs in LD with MS risk SNPs. The human miRNAs proximal to MS risk SNPs and those in eQTL with MS risk SNPs were termed human MS risk miRNAs. Proximal miRNAs were determined using hg19 genome annotation and bedtools v2.26.0 [27]. The eQTL data for miRNA (mir-eQTL) was obtained from the Geuvadis dataset [28].

Putative genomic coordinates of binding regions of host and EBV miRNA were obtained from PAR-CLIP data for LCL cell lines [19]. LCL cell lines were EF3D-AGO2, LCL35, and LCL-BAC (transformed with the plasmid which contains EBV-B95-8 genome), LCL-BAC-D1, LCL-BAC-D2, and LCL-BAC-D3, with the last three cell lines lacking *miR-BHRF1-1*, *miR-BHRF1-2*, or *miR-BHRF1-3*, respectively.

The host mRNA RNA-seq based gene expression data for 373 LCLs derived from the European population were obtained from Geuvadis [28]. The EBV miRNA expression data were obtained from our previous study [24] which was originally generated using Geuvadis small RNA-seq raw data [28]. Additionally, 5 matching donor B cells and LCLs RNAseq data were acquired from GSE126379 [14]. The eQTL data for MS risk SNPs in the LCL context were obtained from our previous study [14] which were originally generated using GTEx 7 dataset [29].

The Regulatory Element Local Intersection (RELI) tool was used to extract binding region data for the EBV transcription factors *BZLF1*, *EBNA2*, *EBNA3c*, *EBNA1* and *EBNALP* [30]. Bedtools v2.26.0 [27] was used to determine the co-localization of MS risk SNPs and EBV transcription factor binding regions.

### 4.2. miRNA Target Site Prediction

Putative EBV miRNA target sites were acquired from the RepTar dataset which is generated by the most efficient algorithm suited for viral miRNA target site prediction on 3′UTR [18]. While 44 EBV derived viral miRNAs have been identified, the EBV B95.8 strain contains only 13 miRNAs due to a large deletion in the *BART* miRNA cluster. Therefore, we identified 13 EBV B95.8 miRNA predicted target sites amongst 223 MS risk genes from the RepTar data set. Next, we filtered out the target sites which passed the threshold of minimal free energy of binding (MFE) ≤ −10, normalized minimal free energy of binding ≥ 0.1, G-U base pairs fraction ≤ 0.25, site conservation ≥ 0 and repeating motifs ≥ 1. Then, LiftOver was used to convert the predicted target site chromosomal positions from hg18 to hg19 [31]. Putative human miRNA (MS risk and non-MS risk) target sites were predicted using TargetScan release 7.2 [16] and miRmap-1.1 [17] which have the highest prediction power amongst the current tools.

### 4.3. EBV and MS Risk miRNA Target Identification (Pipeline A)

Appendix A illustrates the pipeline we developed to identify the EBV and MS risk miRNA targets (pipeline A). Briefly, EBV and host MS risk miRNA targets among human genes were predicted using RepTar [18], TargetScan [16], and miRmap [17]. Then, we identified predicted targets which were verified by PAR-Clip data for EF3D-AGO2, LCL35, LCL-BAC, LCL-BAC-D1, LCL-BAC-D2 and LCL-BAC-D3 using bedtools v2.26.0 [27]. We determined the most accurate miRNA:mRNA interactions were those verified by PAR-Clip data in at least two of three LCL PAR-CLIP datasets with a full EBV 95.8 miRNA profile (EF3D, LCL35 and BAC). PAR-CLIP data for mutant EBV strain cell lines were used for further evaluation of *BHRF1-1*, *BHRF1-2* and *BHRF1-3* targets.

### 4.4. MS Risk SNP Interaction with EBV and Host miRNA Analysis (Pipeline B)

To identify the MS risk SNP effects on miRNA binding, we extended the chromosomal position of the SNPs by 3 base pairs (bps) and 15 bps upstream and downstream, respectively. It has been shown that the sequence and structure within this window flanking the miRNA target site affect miRNA:mRNA duplex formation and stability [32]. Theoretically, a SNP is capable of altering the secondary structure of mRNA within a ± 70 bp window of its position [32,33,34]. Hence, we considered this window as a flanking region for each SNP. Then, we determined the overlaps between the extended genomic coordinates of the MS risk SNP list and PAR-CLIP clusters for EF3D-AGO2, LCL35, LCL-BAC, LCL-BAC-D1, LCL-BAC-D2 and LCL-BAC-D3 using bedtools v2.26.0 [27]. Then, we filtered for hits that were present in at least two of three LCL PAR-CLIP datasets with a full EBV 95.8 miRNA profile (EF3D, LCL35 and BAC). Finally, we predicted the putative miRNAs which target these sequences using TargetScan [16], miRmap [17], and RepTar [18]. Pipeline B is illustrated in Appendix A.

### 4.5. Genotyping and Gene Expression

Blood samples were collected from 52 healthy individuals with informed consent (Westmead Hospital Human Research Ethics Committee Approval 1425, code ETH13877, January 2011). B lymphocytes were purified using immunomagnetic human B cell enrichment Kit (Stem Cell Technologies, Vancouver, BC, Canada) according to the manufacturer’s instructions. For the generation of LCLs, fresh or frozen PBMCs were incubated for 1 h at 37 °C with supernatant from the EBV B95.8 strain cell line. The cells were then suspended in media consisting of RPMI-1640 (Lonza, Basel, Switzerland) supplemented with 10% fetal bovine serum (FBS, Sigma Aldrich, St. Louis, MO, USA), 2 mM L-glutamine (Life Technologies, Carlsbad, CA, USA), 50 units per ml penicillin/50 µg/mL streptomycin (Life Technologies) and 2 μg/mL of cyclosporin A (Sigma Aldrich). The cells were plated at 2.5 × 106 or 5 × 106 cells/well in 48-well plates. Media was supplemented weekly until the cells were expanded into a 25 cm2 flask. LCLs were cryopreserved in 10% DMSO (MP Biomedicals, Santa Ana, CA, USA) 50% FBS and RPMI. All 52 B cells and LCLs were genotyped for rs10271373 using a Taqman SNP genotyping Assay (C_1464338_10) and gene expression for *ZC3HAV1* was assayed using Taqman probe Hs00912661_m1 (Life Technologies) in B cells and LCLs.

### 4.6. Small RNA-Seq

Global miRNA expression profiling using RNAseq was carried out for *n* = 5 LCL and *n* = 5 CD19 + B cells. Total RNA was first isolated using the miRNeasy Mini Kit (Qiagen) before small RNAseq library preparation using the Illumina TruSeq Small RNA Sample Prep Kit. The indexed libraries were pooled and 50 bp single end reads were sequenced on the HiSeq 2500 (Illumina). Reads were assessed for quality using FastQC, aligned to hg19 and EBV reference sequence (NC_007605.1) using TopHat2 [35], and summarised to RPKM gene level expression using SAMmate [36]. Differentially expressed genes were calculated using EdgeR [37].

### 4.7. Statistics Analysis

The Spearman rank-order correlation coefficient test was used to assess the correlation between miRNAs and putative targeted mRNA expression levels [28]. A hypergeometric distribution over-representation test was used to calculate the *p*-value and representation factor for miRNA set overlaps [38]. Analysis of rs10271373 genotype effect on *ZC3HAV1* expression in LCLs and B cells was conducted using GraphPad Prism 9 (GraphPad Software, San Diego, CA, USA) using unpaired (where appropriate) two-tailed t-tests to compare between homozygous risk genotype (AA) and protective allele carrier (heterozygous genotype AC and homozygous protective genotype) groups. Benjamini–Hochberg correction was used to adjust *p* values for multiple comparisons.

## Figures and Tables

**Figure 1 ijms-22-02927-f001:**
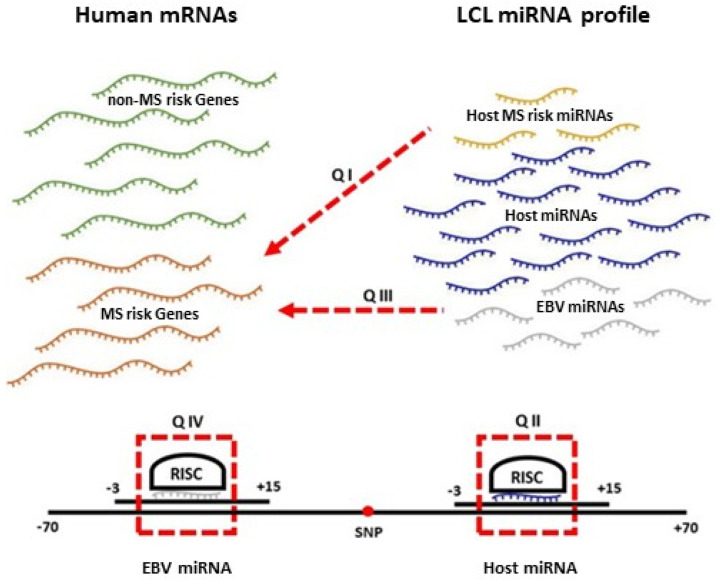
The possible interactions between LCL miRNAs and host/EBV mRNAs which could be involved in MS pathogenesis: (I) Host MS risk miRNAs may target MS risk genes (II) host miRNA binding sites among MS risk genes may be affected by MS risk SNPs (III) EBV miRNAs may target MS risk genes (IV) EBV miRNA binding sites among host MS risk genes may be affected by MS risk SNP.

**Figure 2 ijms-22-02927-f002:**
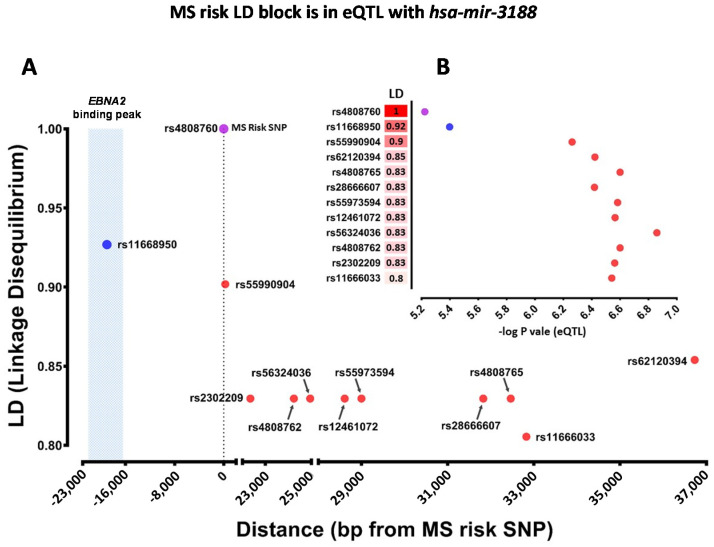
The eQTL effect and architecture of the LD block containing MS risk SNP rs4808760 LD block. rs4808760 is an MS risk SNP which is proximal to *hsa-mir-3188-3p*. Y and X axes in (**A**) demonstrate LD value and location of each SNP in rs4808760 LD block, respectively. Y and X axes in (**B**) demonstrate LD value and eQTL significance of each SNP in rs4808760 LD block, respectively. Blue circle represents a SNP which is targeted by *EBNA2*. The purple circle shows the MS risk SNP. Red circles show the other SNPs in this LD block which are were in eQTL with *hsa-mir-3188-3p* with FDR less than 5% In the Geuvadis Euro cohort (*n* = 373) [PMID: 24037378] of the Geuvadis study.

**Figure 3 ijms-22-02927-f003:**
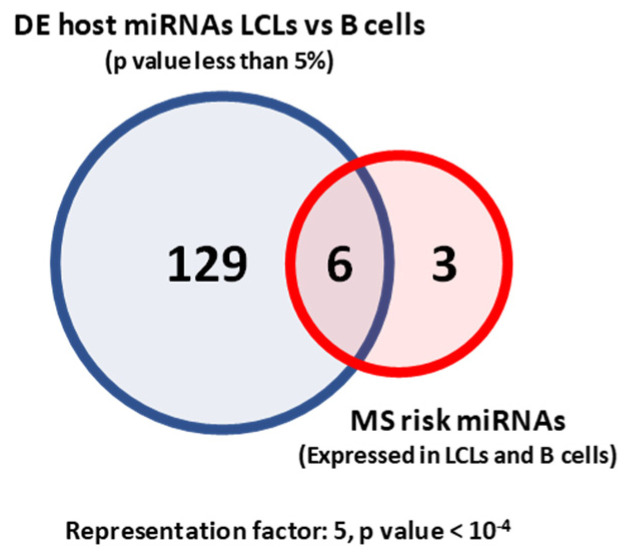
The over-representation of MS risk miRNAs differentially expressed (DE) between LCLs and B cells. We identified differentially expressed miRNAs in LCLs and B cells with a *p* value less than 0.05. MS risk miRNAs over-representation in DE miRNAs between LCLs and B cells.

**Figure 4 ijms-22-02927-f004:**
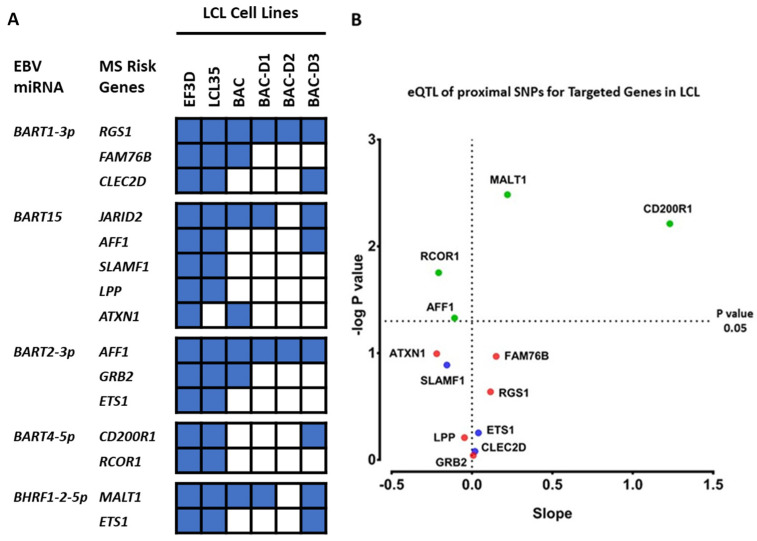
EBV miRNAs’ targets among MS risk genes. (**A**) shows MS risk genes which are targeted by EBV miRNAs and the replication of these interactions in different LCL cell lines (blue fill). LCL cell lines EF3D-AGO2, LCL35 and LCL-BAC (transformed with the plasmid which contains EBV-B95-8 genome) contain complete EBV miRNA profile. LCL-BAC-D1, LCL-BAC-D2 and LCL-BAC-D3 cell lines lacking *miR-BHRF1-1*, *miR-BHRF1-2*, or *miR-BHRF1-3*, respectively. The miRNA:mRNA interactions were deemed to be most accurate if verified by PAR-Clip data in at least two of three LCL PAR-CLIP datasets with a full EBV 95.8 miRNA profile (EF3D, LCL35 and BAC). PAR-CLIP data for mutant EBV strain cell lines were used for further evaluation of *BHRF1-1*, *BHRF1-2*, and *BHRF1-3* targets. (**B**) shows the significance of eQTL effect of MS risk SNPs on genes which targeted by EBV miRNAs. Red dots represent insignificant eQTLs at a cutoff *p* value of 0.05. Blue dots represent the MS risk genes that have an *EBNA2* binding site collocated with the risk SNP. Green dots represent the significant eQTLs at a cutoff *p* value of 0.05 which have an *EBNA2* binding site collocated with the risk SNP. Slope means the effect of risk allele relative to protective allele on expression.

**Figure 5 ijms-22-02927-f005:**
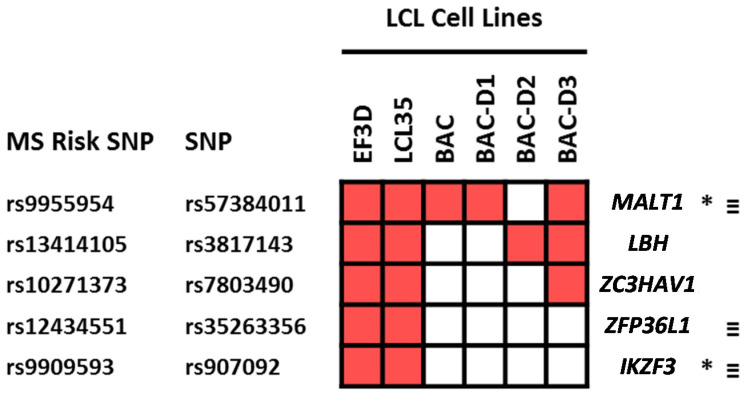
SNPs which are located within or close to miRNA target sites on MS risk genes and the replication of these interactions in the different LCL cell lines used in the PAR-CLIP study (red fill). * shows the genes which are involved in LCL latency III specific MS risk transcriptome. ≡ shows genes where the proximal MS risk SNP, or SNPs in LD with them, are targeted by *EBNA2*.

**Figure 6 ijms-22-02927-f006:**
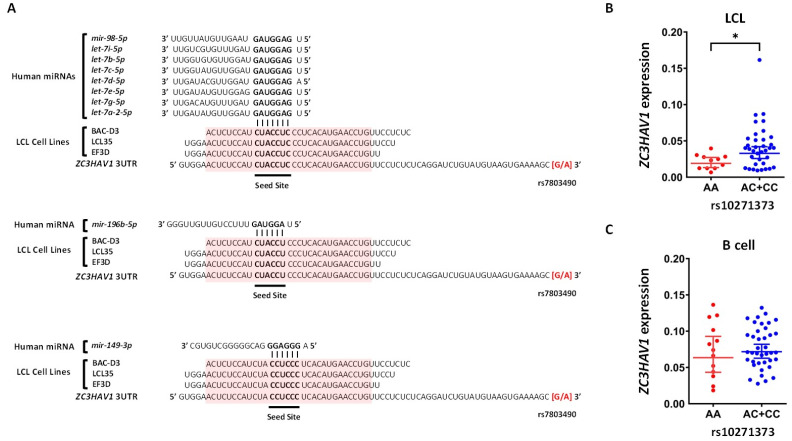
Putative cellular miRNAs target sites on *ZC3HAV1* and the genotype effect of MS risk SNP on *ZC3HAV1* expression in LCLs and B cells. (**A**) The putative cellular miRNAs target sites on *ZC3HAV1* which were located nearby the SNP rs7803490 in LD with MS risk SNP rs10271373. The consensus sequences obtained using pipeline B are highlighted red. The rs10271373 genotype effect on expression of *ZC3HAV1* in LCLs (**B**) and B cells (**C**). AA is the risk genotype of rs10271373, and CC is the protective genotype. * indicates *p* ≤ 0.01.

**Table 1 ijms-22-02927-t001:** MS risk miRNAs.

MS Risk SNP	miRNA Gene Name	Status	Mature miRNA
rs6589939	*MIR100HG*	Proximal	*hsa-miR-125b-5p*
			*hsa-miR-125b-1-3p*
			*hsa-let-7a-5p*
			*hsa-let-7a-2-3p*
			*hsa-miR-100-5p*
			*hsa-miR-100-3p*
rs13385171	*MIR4778*	Proximal	*hsa-miR-4778-5p*
			*hsa-miR-4778-3p*
rs12971909	*MIR4746*	Proximal	*hsa-miR-4746-5p*
			*hsa-miR-4746-3p*
rs34026809	*MIR6716*	Proximal	*hsa-miR-6716-5p*
			*hsa-miR-6716-3p*
rs2150879	*MIR21*	Proximal	*hsa-miR-21-5p*
			*hsa-miR-21-3p*
rs71252597	*MIR4435-1HG*	Proximal	*hsa-miR-4435*
rs6990534	*MIR1204*	Proximal	*hsa-miR-1204*
rs7819665	*MIR1208*	Proximal	*hsa-miR-1208*
rs77654077	*MIR623*	Proximal	*hsa-miR-623*
rs149114341	*MIR4492*	Proximal	*hsa-miR-4492*
rs4808760	*MIR3188*	eQTL	*hsa-miR-3188*

**Table 2 ijms-22-02927-t002:** MS risk miRNA expressions in LCLs compared to B cells.

MS risk miRNA	Expression Status
*hsa-miR-100-5p*	Expressed in LCLs and B cells
*hsa-miR-3188*	
*hsa-miR-21-3p*	Upregulated in LCLs compared to B cells
*hsa-miR-21-5p*	
*hsa-miR-4746-5p*	
*hsa-let-7b-5p*	Downregulated in LCLs compared to B cells
*hsa-let-7e-5p*	
*hsa-miR-4435*	Only expressed in B cells
*hsa-miR-4492*	Only expressed in LCLs
*hsa-miR-125b-1-3p*	Not expressed in LCLs or B cells
*hsa-let-7a-2-3p*	
*hsa-miR-100-3p*	
*hsa-miR-4778-5p*	
*hsa-miR-4778-3p*	
*hsa-miR-4746-3p*	
*hsa-miR-6716-5p*	
*hsa-miR-6716-3p*	
*hsa-miR-1204*	
*hsa-miR-1208*	
*hsa-miR-623*	

**Table 3 ijms-22-02927-t003:** Number of targeted MS risk genes by MS risk miRNAs.

MS Risk miRNAs	Number of Targeted Genes	Number of Targeted MS Risk Genes
*hsa-mir-3188-3p*	35	None
*hsa-miR-21-3p*	67	None
*hsa-miR-21-5p*	11	5
*hsa-miR-125b-5p*	18	4
*hsa-let-7a-5p*	73	1

**Table 4 ijms-22-02927-t004:** MS risk SNP genotype effects on miRNA:mRNA interactions.

MS Risk SNP	miRNA	Targeted Gene	Genotype Condition	Correlation Significance
rs6589939	*hsa-let-7a-5p*	*MAVS*	Whole population	Rho = −0.16, *p* value 0.0005, FDR 0.02
		*END4*	Protective	Rho = −0.18, *p* value 0.03, FDR 0.37
		*ARID3A*	Protective	Rho = −0.18, *p* value 0.02, FDR 0.37
		*USP38*	Protective	Rho = −0.21, *p* value 0.01, FDR 0.27
rs6589939	*hsa-miR-125b-1-5p*	*CD74*	Whole population	Rho = −0.09, *p* value 0.04, FDR 0.32
		*MAP3K10*	Protective	Rho = −0.17, *p* value 0.04, FDR 0.38
rs2150879	*hsa-miR-21-5p*	*MKNK2*	Risk	Rho = −0.27, *p* value 0.02, FDR 0.63
		*TBL1XR1*	Risk	Rho = −0.28, *p* value 0.02, FDR 0.63
		*ALDH9A1*	Risk	Rho = −0.24, *p* value 0.04, FDR 0.77
		*CPEB3*	Risk	Rho = −0.29, *p* value 0.01. FDR 0.63
		*KLHL15*	Risk	Rho = −0.33, *p* value 0.006, FDR 0.63
		*ZFP36L1*	Whole population	Rho = −0.13, *p* value 0.003, FDR 0.06
rs4808760	*hsa-miR-3188-3p*	*PPP2R4*	Risk	Rho = −0.18, *p* value 0.01, FDR 0.63
		*MCL1*	Protective	Rho = −0.49, *p* value 0.01, FDR 0.27
		*TSPYL4*	Protective	Rho = −0.42, *p* value 0.03, FDR 0.37
		*DNAH10OS*	Protective	Rho = −0.4, *p* value 0.04, FDR 0.38
rs2150879	*hsa-miR-21-3p*	*CNIH*	Whole population	Rho = −0.11, *p* value 0.01, FDR 0.11
		*UPF1*	Protective	Rho = −0.21, *p* value 0.02, FDR 0.37
		*DDAH1*	Risk	Rho = −0.26, *p* value 0.02, FDR 0.63
		*ZC3H12C*	Whole population	Rho = −0.1, *p* value 0.02, FDR 0.19

## Data Availability

The data underlying this article are available at the Appendix A.

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
