# Peer review of "The Interaction of Human and Epstein–Barr Virus miRNAs with Multiple Sclerosis Risk Loci"

_ijms, 2021, doi:10.3390/ijms22062927_

Round 1

Reviewer 1 Report

The manuscript entitled "The Interaction of Human and Epstein-Barr virus miRNAs with Multiple Sclerosis risk loci" is fascinating and possesses merit for publication. The study is novel, properly designed, well written, and the results support conclusions. The present manuscript needs to be scan for minor grammatical errors. Additionally, repetition of the text in the Method section "Genotyping and Gene Expression" (line #425- 433) and "Small RNA-seq" (line # 435-442) should be fixed before publication.

Author Response

Response to Reviewer 1 Comments

Q1: The manuscript entitled "The Interaction of Human and Epstein-Barr virus miRNAs with Multiple Sclerosis risk loci" is fascinating and possesses merit for publication. The study is novel, properly designed, well written, and the results support conclusions. The present manuscript needs to be scan for minor grammatical errors. Additionally, repetition of the text in the Method section "Genotyping and Gene Expression" (line #425- 433) and "Small RNA-seq" (line # 435-442) should be fixed before publication.

Answer: Thank you sincerely for reviewing our manuscript, as suggested we have removed the repetition in lines #425-433 and added the correct description for "Genotyping and Gene Expression" to lines #444-457.

Reviewer 2 Report

The concerns are as follows:

  1. The authors perform many comparisons, but they do not address how they deal with the issue of multiple comparisons of multi-dimensional data from small datasets (nearly 5000 SNPs in 223 genes). Please provide a clear statement about whether the p-values are adjusted for multiple comparisons, and how this was done
  2. While such bioinformatics analyses are useful for exploring biological hypotheses, the authors should explain whether they filtered the GWAS SNPs in any away. Specifically, most GWAS SNPs are not causal and it is not reasonable to assume that the SNPs are functional, i.e., will affect mRNA- can they clarify whether for the SNPs selected, such assumptions are reasonable.
  3. In Table 2 – please improve the annotation for the results. For example, the column headed “Expression status”, please clarify whether this is relative to the other cell type and which cell is being compared to which one. For example, the results for has-let-7b-5p indicate that the expression status is “down regulated in LCLs”, but the text results indicate a comparison of those results with B cells. It is unclear from this table whether downregulation in LCLs is relative to no expression, normal expression, or increased expression in B cells. If possible, add quantitative data for this table.
  4. In the results in line 114 – a p<10-4 is provided. Is this an adjusted or crude p value?
  5. In the results looking at EBV miRNAs and MS risk genes – please provide more information about what was considered a significant result. Is it any replication, replication in all cell lines, or at least 2. Why were these cell lines selected?
  6. Please clarify the caption for Fig 4B about green and blue dots – the text is exactly the same “Green dots represent the MS risk genes that have an EBNA2 binding site collocated with the risk SNP” and “ Blue dots represent the MS risk genes that have an EBNA2 binding site collocated with the risk SNP”
  7. In the results on line 200-201 – the authors provide results for LCLs but not B cells. Wouldn’t it be more appropriate to report results for B cells given that they reported miRNA:mRNA interaction in B cells, while those in LCLs were apparently interfered with (Table S5)
  8. Line 214 – is another example of a “significant result” but associated with weak correlations and weak p values. A clearer explanation of how multiple comparisons were dealt with would be helpful.

Author Response

Response to Reviewer 2 Comments

Thank you sincerely for reviewing our manuscript.

Q1: The authors perform many comparisons, but they do not address how they deal with the issue of multiple comparisons of multi-dimensional data from small datasets (nearly 5000 SNPs in 223 genes). Please provide a clear statement about whether the p-values are adjusted for multiple comparisons, and how this was done. and Q8: Line 214 – is another example of a “significant result” but associated with weak correlations and weak p values. A clearer explanation of how multiple comparisons were dealt with would be helpful.

Answer: We agree with reviewer 2 that correction for multiple testing is necessary for this type of analysis. We have updated the manuscript to include Benjamini-Hochberg FDR values to the analyses where multiple comparisons were performed: lines 141, 155, 174, 175, 214, 219, 221, 224, 225 and 228. We also added our approach for adjusting p values for multiple comparisons testing to method section (lines 482-483). Whilst some of the multiple comparisons have resulted in relatively high FDR values, the statistical tests were performed to rank associations from highest to lowest and do reveal that in some instances the associations are not strong.

Q2: While such bioinformatics analyses are useful for exploring biological hypotheses, the authors should explain whether they filtered the GWAS SNPs in any away. Specifically, most GWAS SNPs are not causal and it is not reasonable to assume that the SNPs are functional, i.e., will affect mRNA- can they clarify whether for the SNPs selected, such assumptions are reasonable.

Answer: We agree with reviewer 2’s comment. The majority of the MS GWAS risk SNPs are intergenic, meaning they are not directly affecting the coding region of genes (stop codons, missense mutations etc) but are more likely to be involved in regulation of gene expression (affecting of transcription factor binding, promoter, enhancer etc). For this reason, we have included all SNPs that have reached genome wide significance (5 x 10−8) in the latest MS GWAS meta-analysis (i.e. no additional filtering). We then identified the proximal genes to these GWAS SNPs. We added this explanation to method section (lines 375-379).

Q3: In Table 2 – please improve the annotation for the results. For example, the column headed “Expression status”, please clarify whether this is relative to the other cell type and which cell is being compared to which one. For example, the results for has-let-7b-5p indicate that the expression status is “down regulated in LCLs”, but the text results indicate a comparison of those results with B cells. It is unclear from this table whether downregulation in LCLs is relative to no expression, normal expression, or increased expression in B cells. If possible, add quantitative data for this table.

Answer: Table 2 indicates the MS risk miRNA expressions in LCLs compared to B cells. We added “LCLs compared to B cells” for upregulated and downregulated MS risk miRNAs to Table 2 (lines 111-112).

Q4: In the results in line 114 – a p<10-4 is provided. Is this an adjusted or crude p value?

Answer: This is a crude p value. This p value is generated for the enrichment of MS risk miRNAs among differentially expressed miRNAs between LCLs and B cells using the hypergeometric test. As it is a standalone test there is no correction for multiple testing needed.

Q5: In the results looking at EBV miRNAs and MS risk genes – please provide more information about what was considered a significant result. Is it any replication, replication in all cell lines, or at least 2. Why were these cell lines selected?

Answer: LCL cell lines EF3D-AGO2, LCL35, and LCL-BAC (transformed with the plasmid which contains EBV-B95-8 genome) contain complete EBV miRNA profile. LCL-BAC-D1, LCL-BAC-D2, and LCL-BAC-D3 cell lines lacking miR-BHRF1-1, miR-BHRF1-2, or miR-BHRF1-3, respectively. We determined the most accurate miRNA:mRNA interactions if were verified by PAR-Clip data in at least two of three LCL PAR-CLIP datasets with a full EBV 95.8 miRNA profile (EF3D, LCL35, and BAC). PAR-CLIP data for mutant EBV strain cell lines were used for further evaluation of BHRF1-1, BHRF1-2, and BHRF1-3 targets. As suggested, we have updated this section in the methods with a clearer explanation (lines 415-428). We also updated the caption for Figure 4 (line 178) to improve clarity.

Q6: Please clarify the caption for Fig 4B about green and blue dots – the text is exactly the same “Green dots represent the MS risk genes that have an EBNA2 binding site collocated with the risk SNP” and “ Blue dots represent the MS risk genes that have an EBNA2 binding site collocated with the risk SNP”

Answer: We thank the reviewer for picking up this error. Green dots represent the significant eQTLs at a cutoff p value of 0.05 which have an EBNA2 binding site collocated with the associated risk SNP. We have updated this in the caption for Figure 4b (line 178).

Q7: In the results on line 200-201 – the authors provide results for LCLs but not B cells. Wouldn’t it be more appropriate to report results for B cells given that they reported miRNA:mRNA interaction in B cells, while those in LCLs were apparently interfered with (Table S5).

Answer: We agree that it is appropriate to report results for both B cells and LCLs and this information is available in Table S10 as referred to in the main text.